evolution, microbiology, ecology

*Pseudomonas aeruginosa*, bacteriocins, competition, genetic distance, metabolic dissimilarity

**Author for correspondence:**
Aubrey A. Mojesky
e-mail: aubrey.mojesky@gmail.com

# Spatial structure maintains diversity of pyocin inhibition in household *Pseudomonas aeruginosa*

Aubrey A. Mojesky[1] and Susanna K. Remold[1,2]

[1]Department of Biology, University of Louisville, Louisville, KY, USA
[2]Department of Biological Sciences, University of Massachusetts, Lowell, MA, USA

AAM, 0000-0002-2578-4513

Nearly all bacteria produce narrow-spectrum antibiotics called bacteriocins. Studies have shown that bacteriocins can mediate microbial interactions, but the mechanisms underlying patterns of inhibition are less well understood. We assembled a spatially structured collection of isolates of *Pseudomonas aeruginosa* from bathroom and kitchen sink drains in nine households. Growth inhibition of these *P. aeruginosa* by bacteriocins, known as pyocins in this species, was measured using pairwise inhibition assays. Carbon source usage of these isolates was measured, and genetic distance was estimated using multilocus sequencing. We found that as the distance between sites of isolation increased, there was a significantly higher probability of inhibition, and that pyocin inhibition and susceptibility vary greatly among isolates collected from different houses. We also detected support for other mechanisms influencing diversity: inhibition outcomes were influenced by the type of drain from which isolates were collected, and while we found no indication that carbon source utilization influences inhibition, inhibition was favoured at an intermediate genetic distance. Overall, these results suggest that the combined effects of dispersal limitation among sites and competitive exclusion within them maintain diversity in pyocin inhibition and susceptibility phenotypes, and that additional processes such as local adaptation and effects of phylogenetic distance could further contribute to spatial variability.

## 1. Introduction

A central goal of evolutionary biology is understanding the underlying mechanisms that generate and maintain diversity. One factor thought to promote diversity in microorganisms is the production of bacteriocins, which are narrow-spectrum antimicrobials produced by nearly all species of bacteria. These toxins differ from traditional antibiotics in that their targets are presumably restricted to members of the same species or closely related species, although there is some diversity among the phylogenetic killing range of these toxins [1–3]. The bacteriocins of *Pseudomonas aeruginosa*, called pyocins, are unique in both their highly abundant production and diversity. For example, *P. aeruginosa* has the ability to produce three different types of pyocins: S, R and F. S-type pyocins are similar to the bacteriocins of *Escherichia coli*, called colicins, and are typically composed of a toxin and immunity gene. On the other hand, R- and F-type pyocins resemble phage tails and resistance is thought to be mediated through receptor incompatibility [4]. Furthermore, each of these classes has multiple subtypes and isolates have the ability to produce unique combinations of each type [4–6]. We use inhibition assays to expose the potential pyocin-mediated interactions among environmental isolates of *P. aeruginosa* that may or may not come into contact in the field. The observation of inhibition in these assays can be a result of the production of any type of pyocin

(S, R and/or F) and indicates that processes occurring in the environment contribute to the maintenance of diversity in pyocin inhibition and susceptibility.

One process believed to maintain diversity in bacteriocins is the presence of spatial structure in the environment, particularly when there is limited dispersal. We investigate the capacity to inhibit among *P. aeruginosa* isolates collected at varying levels of spatial separation: isolates from the same drain in a human house, from different drains from the same house, and from different houses in a metropolitan region. We predict a pattern of increased inhibition outcomes as the sites of isolation become more spatial distant, based on likely dispersal limitations among these drains and the large diversity of pyocins described among *P. aeruginosa*, including among household isolates [3,6]. We also expect little inhibition among isolates collected within the same drain, due to the elimination of susceptible phenotypes by pyocin producers or the evolution of resistance to pyocins encountered at the within-drain level. Furthermore, we predict variability in inhibition across different houses due to the stochastic nature of dispersal among sites, combined with the fact that there are no universal optimal pyocin gene contents [7].

Another factor that can promote diversity at a global scale is local adaptation to different environments [8]. This study focuses on isolates from household bathroom and kitchen sink drains, as previous sampling studies have revealed that this is where *P. aeruginosa* is most common in the house [8,9]. Previous sampling studies have shown conflicting results regarding the differences in *P. aeruginosa* recovery between bathroom and kitchen sink drains, with some studies showing *P. aeruginosa* to be more common in bathrooms [8,10] while others show no differences in recovery between drains from these types of rooms [9,11]. This possible difference in recovery draws attention to the fact that environments may have differences such as the physical shape, frequency of use and types of inputs associated with them. Such differences may result in differences in the relative contributions of competition versus selection by the environment. In the context of different types of household drains, this might result in differential exposure to one another and selection to inhibit one another, leading ultimately to differences among drain types in their inhabitants' inhibition profiles. We evaluated the relationship between drain type and recovery rate and inhibition patterns to investigate potential contributions of local adaptation to different habitat associated with bathroom versus kitchen sink drains.

Although there are indications that bacteriocin-mediated inhibition may not be directly predictive of competitive outcomes [6], there are two well studied hypotheses regarding the relationships between bacteria and whether or not bacteriocin-mediated inhibition should be favoured [12–14]. First, if bacteriocins do contribute to the outcome of competitive interactions, it may be that bacteriocin-mediated inhibition will occur when two individuals have similar resource requirements, given that competition occurs over shared resources [12–15]. This is because while costly production of bacteriocins is not favoured when niche overlap is minimal, it becomes more beneficial as niche overlap increases. As niche overlap becomes even stronger, the capacity to inhibit the competitor may decrease not because there is insufficient benefit to bacteriocin-mediated inhibition, but because the competitors may be similar in their bacteriocin genotypes as well as their metabolic profiles, causing them to lack the capacity to inhibit one

another [12,14]. As such, we predict a 'hump-shaped' relationship between inhibition and metabolic dissimilarity in which inhibition is least favoured among isolates with low and high resource overlap and is most likely to occur when interacting isolates share an intermediate number of resources. Previous studies have examined this relationship between resource competition and bacteriocin-mediated interactions but results vary in the degree to which they conform to this prediction [12–14].

Second, a hump-shaped relationship such that bacteriocin-mediated inhibition is maximized when two individuals are genetically related at an intermediate level [14] is also predicted. This is because bacteriocin production targeting very closely related isolates will not be favoured, because shared phylogenetic history will cause closely related isolates to carry the same bacteriocin immunity genes and therefore not be killed by their shared toxins. Bacteriocin-mediated competition should also be minimal when two individuals are genetically distant if, along with other traits, their resource use overlap declines over evolutionary time. The hump-shaped relationship has been found among laboratory and clinical isolates of *P. aeruginosa* by Schoustra *et al.* [14], but other studies of environmental microbes have found contrasting evidence for this relationship [12,13,16].

The goal of this study was to elucidate the underlying mechanisms that drive diversity in bacteriocin-mediated inhibition. Relationships among spatial scale, metabolic dissimilarity, genetic distance and bacteriocin inhibition have been examined in different microbial systems with differing results. Furthermore, these patterns have not yet been studied in household isolates of *P. aeruginosa*, where the high production of pyocins [3,6] suggests an important ecological role for these protein structures. We found the lowest probability of inhibition to be among isolates collected from the same drain and the highest probability of inhibition to be among isolates collected from different houses, and we identified additional substantial random variation among specific pairs of houses. No relationship was found between metabolic dissimilarity and inhibition; however, a significant hump-shaped relationship was found between genetic distance and inhibition, meaning that isolates of an intermediate genetic distance displayed the highest probability of inhibition.

## 2. Methods

### (a) Bacterial isolate sampling

Seventy houses in the Louisville, Kentucky metropolitan area were sampled from June 2016 to May 2017. Bathroom and kitchen sink drains were sampled using a sterile cell scraper. Samples from all 70 houses were collected in sterile 1X PBS and stored at 4°C. Samples were plated onto cetrimide agar [17] and incubated at 37°C for 24–48 h. If growth was observed on the plate, colonies were isolated and species-specific PCR primers were used to identify those colonies that were *P. aeruginosa* [18].

For the first 12 houses, up to 13 clonal picks were performed for each drain sample to maximize likelihood of recovery. Based on results of this sampling, a standardized protocol was used in sampling of houses 13–70. In these latter houses, the minimum of all colonies on a plate or up to seven colonies were isolated and *P. aeruginosa* was identified using a species-specific PCR reaction [18].

Across all 70 houses, nine houses yielded a total of 105 isolates. All *P. aeruginosa* isolated (105 isolates in all) were stored in 20% glycerol at −80°C. Of these, three isolates were randomly

chosen from each of the two drain types (bathroom sink drain and kitchen sink drain) in all nine houses to be used in pairwise inhibition assays. This resulted in a total of 54 *P. aeruginosa* isolates for the pairwise inhibition assays.

## (b) Pairwise inhibition assays

The 54 (three isolates × two drain types × nine houses) isolates were tested for ability to inhibit and resist inhibition in all pairwise combinations according to the protocol described by Fyfe *et al.* [19]. Each pair was assayed three independent times, for a total of 2916 tests of inhibition.

Briefly, individual clones of the 54 isolates were picked and used to inoculate 4 ml of LB broth. Each culture was grown for about 18 h at 28°C with shaking at 250 r.p.m. After the initial incubation period, 10 µl of this overnight culture was used to inoculate two tubes of 4 ml LB broth. Cultures were incubated with shaking at 28°C for 2–3 h. Pyocin production was then induced in one culture (the 'producer' culture) using mitomycin C, a DNA damaging agent that has been shown to induce bacteriocin expression [1,20], to a final concentration of $2 \, \mu g \, \mu l^{-1}$. The second, uninduced ('indicator') culture was also incubated alongside the producer culture.

After 16–18 h incubation at 28°C, 5 µl of the producer cultures were spotted onto LB plates and incubated for 6 h at 28°C. 1.5 ml of chloroform was added to each plate to kill the producer colonies and plates were allowed to dry for 30 min. Producer colonies were killed prior to soft agar overlay to avoid the observation of inhibition due to a mechanism other than pyocin production, such as contact-dependent or secretion-mediated killing. In each experimental block, the indicator cultures were standardized to the lowest observed $OD_{595}$ reading among the set of 54. Once standardized in this way, 100 µl of each indicator culture was used to inoculate 6 ml of soft agar, which was vortexed gently and poured over the agar surface of plates that had been spotted with killed producer cultures. Plates were left to incubate for 16–18 h at 28°C. Zones of inhibition of the indicator isolates were identified by visual inspection. Cases of inhibition presented as either a clear, defined zone of inhibition (indicating R or F-type pyocin) or diffuse zone of inhibition (indicating S-type pyocin inhibition); both inhibition phenotypes were considered cases of inhibition for the purposes of data collection.

## (c) Genetic distance

Genomic DNA was extracted from isolates using 1 ml of overnight culture and Wizard Genomic DNA purification kit (Promega, Madison, WI, USA). Genetic distances were estimated using Illumina whole genome sequencing and ARIBA [21] which was used to extract seven housekeeping genes (*ascA*, *aroE*, *guaA*, *trpE*, *mutL*, *nuoD*, *ppsA*) used in a *P. aeruginosa* multilocus sequence typing scheme developed by Curran *et al.* [22]. These genes were concatenated and pairwise genetic distances were calculated using only the seven housekeeping gene sequences and R package ape [23] on RStudio v. 1.1.453 [24].

## (d) Resource competition

The ability of the 54 *P. aeruginosa* isolates to use 31 carbon sources was measured in triplicate using EcoPlates (Biolog, Hayward, CA, USA) and a protocol adapted from Schoustra *et al.* [14]. Cultures were grown overnight to stationary phase to a minimum $OD_{595}$ of 0.8. Cultures were diluted by adding 10 µl of overnight culture to 10 ml of minimal salt media ($N_2HPO_4$ 6.8 g, $KH_2PO_4$ 3 g, NaCl 0.5 g, $NH_4Cl$ 1 g, 1000 ml $dH_2O$). After a starvation period of two hours, plates were inoculated with 150 µl of sample. The $OD_{595}$ of all wells in the plate were measured immediately after inoculation ($T_0$) and again after 48 h of incubation at 28°C. The absorbance measures from the 48-hour time point were used to calculate pairwise Bray–Curtis dissimilarity indices with R package vegan [25].

## (e) Statistical analysis

To assess the differences in recovery from bathroom and kitchen sink drains, presence or absence of *P. aeruginosa* from each drain was analysed using McNemar's test in the set of 58 houses which were sampled using a standardized sampling technique.

Four generalized linear mixed models were performed with the SAS v. 9.4 TS1M3 GLIMMIX procedure [26]. Models were run with LaPlace estimation and containment method of denominator degrees of freedom estimation. Self-on-self interactions were excluded from all analyses, as no inhibition was expected or observed in these controls. The results of the three replicate assays per pair were reduced to a dichotomous response variable, inhibition outcome by using the most common (two out of three) outcome when all three did not agree.

We examined the importance of spatial scale by running a generalized linear mixed model with inhibition as a binary response variable and spatial scale as a categorical predictor variable. House of isolation was not included in this model as a predictor variable because it is confounded with spatial scale, but was evaluated in all other models. This was done by including the house of the producer, the house of the indicator, as well as the interaction between the two as random effects. We also tested the effect of drain type by running a generalized linear mixed model with inhibition as a response variable and producer drain type, indicator drain type, and the interaction between the two terms included as fixed predictor variables.

We explored the hump-shaped relationship between metabolic dissimilarity and inhibition by running a generalized linear mixed model with metabolic dissimilarity and the square of metabolic dissimilarity as predictor variables and call as the response variable. Similarly, we ran a generalized linear mixed model using genetic distance and the square of genetic distance as predictor variables and call on inhibition as the response variable.

# 3. Results

## (a) *Pseudomonas aeruginosa* is rarely found in multiple drains within a house

The bathroom and kitchen sink drains of 70 houses were sampled in the Louisville metropolitan area. 58 of these 70 houses were sampled using a standardized sampling protocol; this subset of houses was used for analysis of *P. aeruginosa* recovery. Of these 58 houses, 22 yielded *P. aeruginosa* from at least one drain with an overall recovery of roughly 37%. *Pseudomonas aeruginosa* was collected from the bathroom sink drain only in 11 houses (19%), from the kitchen sink in only five houses (9%) and from both the bathroom and kitchen sink drain in six houses (10%). Despite slightly higher recovery from bathroom sink drains than from kitchen sink drains, this analysis identified no significant difference in recovery from the two drain types ($p = 0.21$, McNemar's test).

## (b) Inhibition varies across space

We tested an isolate's ability to inhibit using an agar overlay method from a well-established protocol [19], which uses the DNA damaging agent mitomycin C, to induce bacteriocin expression. Pairwise interactions were conducted, meaning that each isolate was examined as both a 'producer' and as an 'indicator'. As a producer, an isolate was treated with mitomycin C and was tested for its ability to kill a non-induced

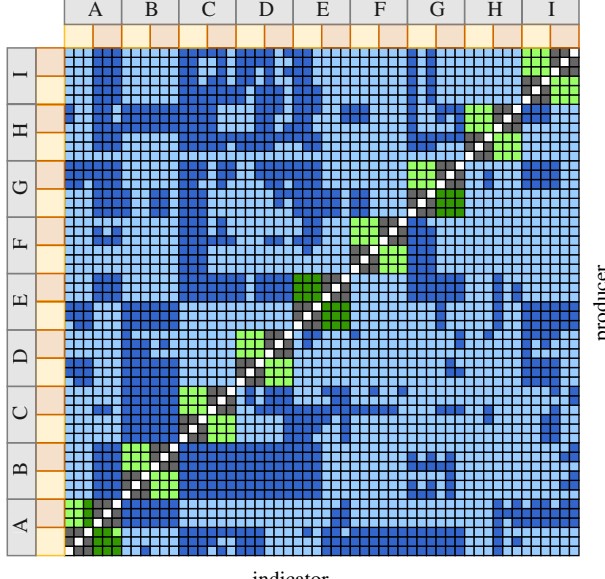

**Figure 1.** Pairwise inhibition assay outcomes. Light grey boxes with letters on both axes represent the nine houses from which bathroom and kitchen sink isolates were collected. Yellow and orange boxes represent isolates obtained from bathroom or kitchen sink drains, respectively. Pairwise interaction assay results were colour coded based on a call on the outcome of inhibition, which was made if two out of three interactions resulted in inhibition. White squares represent no self-inhibition. Dark blue squares represent inhibition among isolates collected from different houses. Dark green squares represent inhibition among isolates collected from the same house and dark grey squares represent no inhibition observed among isolates collected from the same drain. Cases of non-inhibition across houses are indicated in light blue and cases of non-inhibition between drains of the same house are indicated in light green. (Online version in colour.)

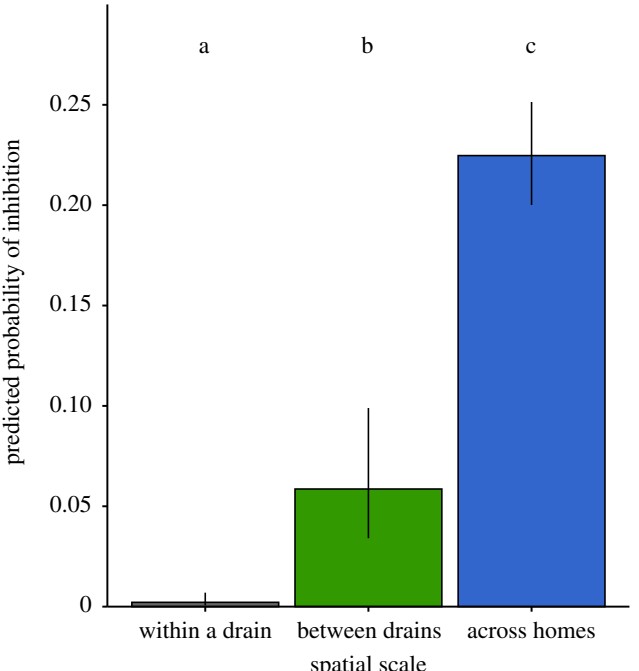

**Figure 2.** Predicted probability of inhibition when isolates are collected at different spatial scales. Least-squares means with 95% confidence intervals display the predicted probability of the producer isolate to inhibit the indicator isolate. Different letters indicate significant differences in means at $\alpha = 0.05$ after correcting for multiple comparisons using the Tukey–Kramer adjustment. (Online version in colour.)

**Table 1.** Mean percentage of inhibition across scale.

| scale | mean percentage of inhibition |
|---|---|
| within a drain | 0 |
| between drains | 23.50% |
| across houses | 34.90% |

strain, termed the indicator. Although the presence of pyocins was not directly confirmed during the inhibition assays, the presence of pyocin genes within the isolates used in these assays was confirmed comparison with the BAGEL database [27] (A.A.M. 2018, unpublished data). Additionally, zones of inhibition observed in the assays presented as diffuse or clear zones of inhibition, and did not present as plaques that would be expected if bacteriophage was responsible for killing. Previous studies have also repeatedly shown that an induction with mitomycin C is a reliable method to induce the expression of pyocin genes [1,20] Additionally, no cases self-inhibition were observed in the assays, suggesting that pyocins indeed were responsible for the zones of inhibition as each isolate should carry an immunity gene to its own toxin [28].

Self-on-self interactions, as expected, never resulted in inhibition (white squares, figure 1). Of the remaining 2862 pairwise interactions, overall, 944 (33%) resulted in inhibition. The interactions in this study were observed across three spatial scales: within the same drain, between different drains in the same house, and across different houses. No inhibition was observed between isolates collected from the same drain (dark grey squares, figure 1). Inhibition was observed among isolates collected from different drains in the same house in three of nine houses (dark green squares). The majority of inhibition was observed among isolates collected from different houses (dark blue squares). A generalized linear mixed model assessing dependence of the outcome of

the inhibition assay on spatial scale showed that as distance increased, there was a significantly higher probability of inhibition (figure 2, $p < 0.0001$; table 1).

We also examined patterns of pyocin inhibition and susceptibility among isolates collected across different houses through the inclusion of house as a predictor variable in generalized linear mixed models with metabolic and genetic distance. We found significant variability attributable to both the house of the producer and the house of the indicator. In other words, whether or not a producer is able to inhibit an indicator depends on the house of isolation for both interacting isolates (figure 1). Therefore, these results suggest that, in addition to scale, there is stochasticity across space that contributes to diversity in pyocin phenotype.

## (c) Inhibition varies with drain type

Previous studies have shown that bathroom and kitchen sink drains often differ in their recovery of *P. aeruginosa* [8,10]. Based on these differences in recovery between drain types, we hypothesized that isolates collected from different drain types may differ with respect to their pyocin inhibition or susceptibility. To address this, pyocin inhibition was measured

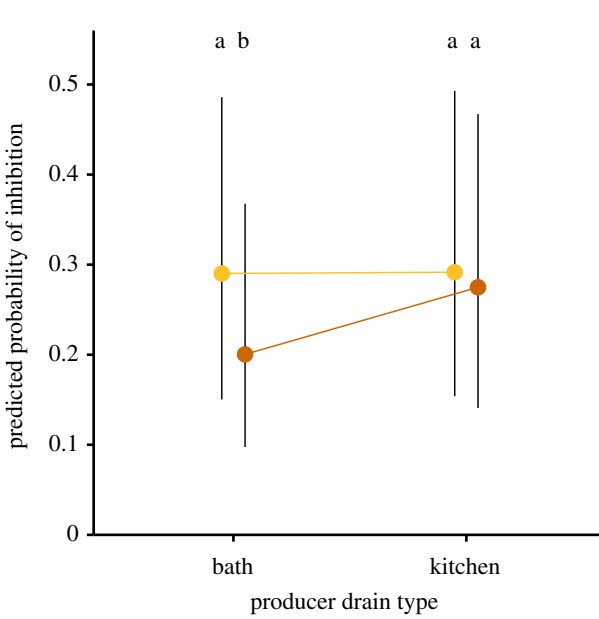

**Figure 3.** Predicted probability of inhibition within isolates collected from different drain types. Predicted probabilities of inhibition are least-squares means with 95% confidence intervals. Different letters indicate significant differences in means at $\alpha = 0.05$ after correcting for multiple comparisons using the Tukey–Kramer adjustment. (Online version in colour.)

**Table 2.** Generalized linear mixed model testing the effect of metabolic dissimilarity and the square of metabolic dissimilarity as well as various random effects on the outcome of inhibition.

| effect | estimate | d.f. | test statistic[a] |
|---|---|---|---|
| metabolic dissimilarity | 2.5926 | 2515 | 0.48 |
| metabolic dissimilarity$^2$ | 2.6765 | 2515 | 0.1 |
| producer house | 0.224 | | 0.76 |
| indicator house | 0.9343 | | 1.47 |
| ordered combination of drain types | 0.03819 | | 0.99 |
| producer house × indicator house | 2.489 | | 3.89*** |

[a]Fixed effect is tested with an approximate $F$-test. Random effects are tested using Wald $Z$-tests.
***$p < 0.001$.

**Table 3.** Generalized linear mixed model testing the effect of genetic distance and the square of genetic distance as well as various random effects on the outcome of inhibition.

| effect | estimate | d.f. | test statistic[a] |
|---|---|---|---|
| genetic distance | 308.18 | 1871 | 17.84*** |
| genetic distance$^2$ | −17 798 | 1871 | 11.97*** |
| producer house | 0.211 | | . |
| indicator house | 0.9595 | | 1.4 |
| ordered combination of drain types | 0.05152 | | . |
| producer house × indicator house | 2.2006 | | 3.30*** |

[a]Fixed effect is tested with an approximate $F$-test. Random effects are tested using Wald $Z$-tests.
***$p < 0.001$.

across two different household drain types: bathroom sink drains and kitchen sink drains. A generalized linear mixed model was run with producer drain type, indicator drain type, and the interaction between these two terms as fixed effects. We found that the inhibition observed was strongly affected by the particular combination of drain types from which the producer and indicator isolates were obtained (interaction effect, $p = 0.05$). In particular, when indicators from kitchen sink drains were challenged with producers from bathroom sink drains, significantly less inhibition occurred in comparison to interactions between any other combination of isolation sources, and none of the other combinations resulted in differences in inhibition (figure 3). The detection of overall greater capacity to inhibit and greater resistance to being inhibited among kitchen sink isolates (producer main effect, $p = 0.0248$, indicator main effect, $p = 0.0026$) is likely driven by this single combination of drain types.

## (d) Metabolic dissimilarity does not predict inhibition

To determine if metabolic dissimilarity is an important predictor of the outcome of inhibition, we measured resource use of 31 different carbon resources using EcoPlates (Biolog, Inc., Hayward, CA, USA) and calculated pairwise metabolic dissimilarities. We then ran a generalized linear mixed model with inhibition as the response variable and metabolic dissimilarity and the square of metabolic dissimilarity as predictor variables (table 2). Neither metabolic dissimilarity nor its quadratic term was found to be significant in the model ($p = 0.4905$ and $0.7486$, respectively). Therefore, we found no evidence of a hump-shaped relationship between metabolic dissimilarity and inhibition. These results indicate that competition mediated by differential use of these common carbon sources may not play an important role in determining the outcomes of inhibitory interactions.

## (e) Genetic distance has a hump-shaped relationship with inhibition

To determine if genetic distance has a hump-shaped relationship with inhibition, we measured genetic distance using differences among isolates' sequences at seven housekeeping genes in the *P. aeruginosa* genome. We looked to see if inhibition occurs most frequently at an intermediate genetic distance by running a generalized linear mixed model with inhibition as the response variable and genetic distance and the square of genetic distance as the predictor variables (table 3). This model identified a significant positive value for the parameter estimating the linear effect of genetic distance ($p < 0.0001$) and fitted a negative parameter estimate for the effect of the square of genetic distance ($p = 0.0006$). These estimates model the predicted hump-shaped relationship for genetic distance and bacteriocin-mediated interactions in which these interactions are predicted to

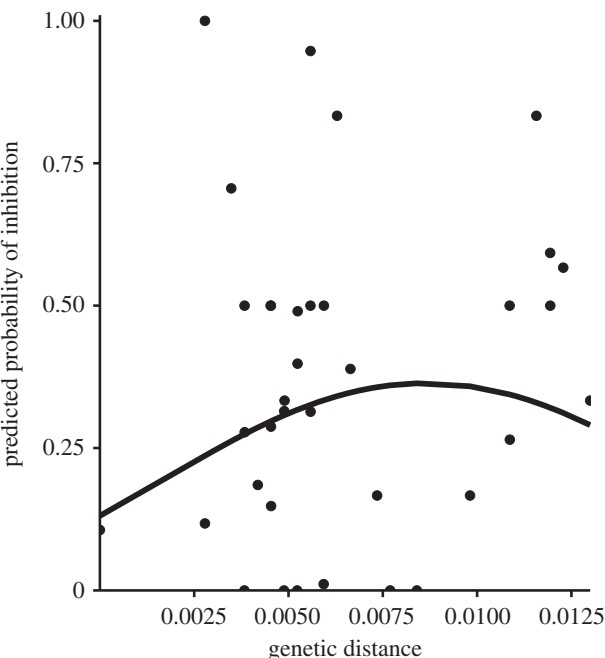

**Figure 4.** Predicted probability of inhibition with genetic distance. The line is plotted using estimates from the generalized linear mixed model run to examine the effect of genetic distance and the square of genetic distance on the outcome of inhibition. Points represent frequency of inhibition for each measure of genetic distance, calculated by averaging call on the outcome of inhibition among all interacting isolates of a particular genetic distance.

occur most frequently when two isolates are intermediately genetically related (figure 4).

## 4. Discussion

### (a) *Pseudomonas aeruginosa* is not ubiquitous in household drains

In this study, we found a greater percentage of drains containing *P. aeruginosa* than in previous sampling studies done by our group (37% as opposed to 15 and 28%) [8,9]. The sensitivity of detection of *P. aeruginosa* was expected to be higher than in similar studies, for two reasons. The culture-based sampling employed the more species-specific medium cetrimide rather than Pseudomonas Isolation Agar [17], and more isolates were collected per sample taken. However, despite our intensive sampling focused on the single type of site in the house where *P. aeruginosa* has previously been shown to most frequently inhabit, and despite higher recovery in this study from each drain type than in those previous studies, we found that the proportion of houses from which *P. aeruginosa* was recovered from both bath and kitchen sink drains was still relatively low (10%). This suggests that, in addition to dispersal into the house, within-house forces such as dispersal limitation or selective differences between bathroom and kitchen sink drains limit the distribution of this species. This is notable in part because it runs counter to the traditional characterization of this species as a ubiquitous generalist [29,30].

### (b) Inhibition increases with scale and varies with house

In our assays, we found that inhibition increases with the separation in the isolation sites of the producer and indicator (figure 2). This suggests that the selective pressure from pyocin producing isolates of *P. aeruginosa* may have eliminated diversity with respect to this phenotype at the within-drain level. Isolates within the same drain have either co-evolved as to not be susceptible to the particular pyocins being produced within a single drain, or pyocin sensitive isolates have been competitively excluded by producer isolates. In contrast, isolates taken from different houses that were less likely to have interacted showed greater diversity in pyocin phenotypes and were able to inhibit one another. The positive relationship among inhibition and increasing spatial scale is consistent with those of previous studies in *Pseudomonas fluorescens* and in species of the *Xenorhabdus* genus [12,16]. Although, this trend was not observed among soil pseudomonads by Kraemer *et al*. [13], the scale on which the isolates were collected in that study may have been too large to detect a relationship with inhibition. Unlike studies done in other *Pseudomonas* species and species in the *Xenorhabdus* genus [12,13,16], we found a much higher frequency of inhibition among isolates overall, which is likely explained by the highly abundant production of pyocins in *P. aeruginosa*, and particularly in environmental isolates of *P. aeruginosa* [3,6].

Previous studies have shown that spatial structure in microbial populations may lead to a decrease in local population diversity, but an increase in overall diversity across larger spatial scales [12,13,16]. In laboratory strains of *Escherichia coli*, it has been demonstrated that in unstructured environments such as a broth culture, bacteriocin production will lead to a decrease in diversity as one strain will competitively exclude the other. However, in a structured environment such as that of an agar plate, competitive interactions occur more locally, which allows for the coexistence of multiple bacteriocin phenotypes [31].

Similarly, Inglis *et al*. [32] also found that a pyocin producing strain of *P. aeruginosa* has increased fitness relative to a non-pyocin producer under conditions of local competition. This is because, when competition is localized, individuals which are genetically related to the bacteriocin producer are close by and are likely to benefit as bacteriocin production frees up resources. High rates of dispersal, however, will cause competition to occur more globally and bacteriocin production will provide less of a fitness benefit. Based on the results of our sampling study, it appears that *P. aeruginosa* dispersal among houses, and, to a lesser extent, within the houses, could potentially be limited. Moreover, even accounting for differences among drain type, metabolic similarity and genetic distance, significant variability in pyocin phenotype was detected among *P. aeruginosa* isolated from different houses. This further supports our conclusion that dispersal limitations across different houses leads to the maintenance of diversity in inhibitory interactions. Further investigation into the relationship between highly patchy environments, like the built environment, and diversity in bacteriocin inhibition in this species and others, is warranted.

### (c) Local adaptation to drain type may influence inhibition

Previous sampling studies have yielded varying results regarding the difference in *P. aeruginosa* recovery from bathroom and kitchen sink drains, with either greater recovery from bathroom drains [8,10] or no differences detected [9,11]; our study showed a non-significant trend toward greater

recovery in bathroom drains. In evaluating possible differences in the outcome of inhibition assays with respect to isolates' house drain types, we observed differences in the pyocin inhibition and sensitivity profiles. We found that inhibition assay outcomes depended on the drain types of interacting isolates. This suggests that local adaptation to drain type may be at play and responsible for differences in the competitive interactions among those isolates.

While we have not explored the particular ecological differences between bathroom and kitchen sink drains, there may be both abiotic and biotic components of these drain types that could contribute to local adaptation of *P. aeruginosa* isolates. For example, there could be differences in available resources or in microbial community compositions, either of which could affect selection through competitive interactions among *P. aeruginosa*. Future studies in this area could work to clarify the differences in drain types with respect to both abiotic and biotic factors.

## (d) Resource competition does not drive inhibition

We looked for differences in resource use among isolates to test the prediction that resource dissimilarity may help predict inhibitory interactions [12–14], but found no evidence of such a relationship. Schoustra *et al.* [14] observed the frequency of inhibition to peak at intermediate resource usage overlap among clinical and laboratory *P. aeruginosa* isolates. Differences between our results and that study could be due the fact that we used only environmental isolates, not clinical isolates and laboratory strains. Kraemer *et al.* [13] found metabolic dissimilarity to negatively correlate with frequency of inhibition, while Bruce *et al.* [12] similarly found that inhibition was significantly higher between isolates with greater niche overlap, suggesting that competition for shared resources may play an important role in the selection for and maintenance of bacteriocin production. This could occur if outcompeting other isolates with similar resource requirements through bacteriocin production liberated limited resources and provided the bacteriocin producing strain with a competitive advantage. The differences between the findings in our study and the outcomes of Bruce *et al.* [12] and Kraemer *et al.* [13] could be attributed to the fact that Kraemer *et al.* [13] were likely examining relationships among different *Pseudomonas* species rather than within one species, and Bruce *et al.* [12] calculated niche overlap as opposed to metabolic dissimilarity as we did. Although we detected no significant relationship between metabolic dissimilarity and pyocin inhibition profile, we note that there are many ways in which resource similarity could differ among isolates that would not be detectable in our assay, particularly in light of the known metabolic versatility of *P. aeruginosa* [29].

Additionally, the carbon resources in Biolog EcoPlates are biased toward plant-based carbon sources, which are probably different from the carbon sources available in sink drains. Future work addressing metabolic similarity could explore other resources over which competition might be occurring such as alternative carbon sources, nitrogen sources, and other bacterial nutritional requirements that might be more available to isolates in sink drains. It can also be noted here that Bara *et al.* [6] found that the ability to inhibit another isolate via pyocin production does not always predict overall competitive ability, so it is possible that pyocin inhibition does not play a critical role in competition and therefore is unrelated to resource use. As such, future investigation into the specific ecological functions of pyocins could further illuminate the maintenance of these antibiotics in microbial populations.

## (e) Inhibition is most favoured at an intermediate genetic distance

We examined the relationship between genetic distance and patterns of inhibition among the isolates in this study and found strong evidence to support a hump-shaped relationship. These results are consistent with the predicted hump-shaped relationship that was observed by Schoustra *et al.* [14], though not with those of Hawlena *et al.* [16], Kraemer *et al.* [13] and Bruce *et al.* [12], which examined such relationships among more distantly related taxa and used different approaches to measuring genetic distance.

Because the measure of genetic distance that we used is based on sequence comparison of conserved housekeeping genes, it measures genetic similarity based on phylogenetic history, and is not mechanistically driven by selection on pyocin use. We know that bacteriocin production in *P. aeruginosa*, particularly in household isolates [6], is much more common than that of other *Pseudomonas* and bacterial species [3]. Thus, the hump-shaped relationship between this form of genetic distance and inhibition patterns could be caused by widespread use of pyocins and restricted movement in space, which shape the population genetic structure of this species. We note that other measures of genetic distance could reveal additional evolutionary dynamics affecting patterns of pyocin interactions. For example, a decline in killing at greater genetic distance can occur if more genetically distant bacteria lack the particular cell surface receptors required for bacteriocin binding [28,33,34]. Such patterns would be better detected using measures of genetic distance based on gene content or gene sequence of pyocin associated genes such as toxin, immunity, regulatory, and receptor synthesis genes. Use of such different genetic distance measures might not reveal the hump-shaped relationship that we detected with our measure.

## 5. Conclusion

In this study, we explored many potential driving forces of diversity in pyocin-mediated inhibition using isolates collected from spatially structured bathroom and kitchen sink drains across nine different houses. Differences in the outcome of inhibition among isolates from different drain types and among isolates differing in their genetic distance suggest roles for local adaptation and/or limited local dispersal, and a signature of phylogenetic history in maintaining pyocin diversity. However, we also found strong patterns indicating an important role for dispersal limitation at a landscape level in driving patterns of pyocin-mediated inhibition. When combined with competitive exclusion within a drain, dispersal limitation can explain an increase in inhibition with spatial scale and random house-to-house variation in patterns of inhibition.

Data accessibility. The dataset and SAS code supporting are available from the Dryad Digital Repository: https://dx.doi.org/10.5061/dryad.w6m905qn0 [35].

Authors' contributions. A.A.M. and S.K.R. designed the study, analysed the data and wrote the manuscript. A.A.M. collected the data. Both authors gave final approval for publication.

**Competing interests.** We declare we have no competing interests.

**Funding.** This work was supported by a National Science Foundation grant (no. DEB-0950361) to S.K.R.

**Acknowledgements.** We acknowledge Zachary Matson, Andrew Diddle and Erica Miller for their help in drain sampling and processing. We thank Tom Hundley for his work on the development of the sampling protocol and Jeff Bara for his guidance on protocols for sampling and inhibition assays. We thank Rachel Whittaker and her group for their whole genome sequencing of the isolates as well in their help in the genome assembly and genetic data analyses. We also thank Debbie Yoder-Himes for her review of this manuscript.

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
