## [Reviewer comments · Proceedings of the Royal Society B: Biological Sciences]

Review History

RSPB-2020-1706.R0 (Original submission)

Review form: Reviewer 1

Recommendation

Accept with minor revision (please list in comments)

Scientific importance: Is the manuscript an original and important contribution to its field?

Good

General interest: Is the paper of sufficient general interest?

Good

Quality of the paper: Is the overall quality of the paper suitable?

Good

Is the length of the paper justified?

Yes

Should the paper be seen by a specialist statistical reviewer?

No

Do you have any concerns about statistical analyses in this paper? If so, please specify them explicitly in your report.

No

It is a condition of publication that authors make their supporting data, code and materials available - either as supplementary material or hosted in an external repository. Please rate, if applicable, the supporting data on the following criteria.

Is it accessible?

Yes

Is it clear?

Yes

Is it adequate?

Yes

Do you have any ethical concerns with this paper?

No

Comments to the Author

In this study, the authors investigate the inhibition/susceptibility profiles of *P. aeruginosa* isolates taken from different drains in different houses. Whilst I started looking at the sampling rationale with some scepticism, having read the ms, I enjoyed the thinking behind it. There is very little that needs improving in this manuscript, beyond a few typos and the comments below:

L178 - The description of the inhibition assay is a little confusing - why are the producer colonies killed prior to overlay, do the producers not need to be alive to interact with the indicator strains? This may be my ignorance, but it is not clear.

L200 - why was more genetic information not used for genetic distance rather than MLSTs. How many housekeeping genes were used?

L389 - Whilst I understand the data used to draw the conclusions about local adaptation, I am unsure whether there is sufficient evidence to suggest that the adaptation is to the drain environment. If this was the case a key hypothesis would be that carbon source utilisation would be significantly different (see next comment), maybe this reduction in inhibition is due to adaptation to the community, rather than the environment. As these communities are stable (a guess?), then there could have been a clonal expansion of a competitive *P. aeruginosa* that has undergone adaptive radiation perhaps...

L406 - I am a big fan of Biolog EcoPlates, however, I'm not sure that these were the best assay to measure the CLPP of these systems as they are biased towards carbon sources derived from plants rather than whatever maybe coming down the bathroom sink. I feel this section and the previous section are intertwined and maybe slimmed and combined.

Figures 3 & 4 are not a big issue, but I find them difficult to interpret as they are not clearly supporting the text. Is there a better way of illustrating this? Perhaps scatterplots with all the data?

Figure 5 is not cited in the text line 325?

Review form: Reviewer 2

Recommendation

Major revision is needed (please make suggestions in comments)

Scientific importance: Is the manuscript an original and important contribution to its field?

Acceptable

General interest: Is the paper of sufficient general interest?

Good

Quality of the paper: Is the overall quality of the paper suitable?

Good

Is the length of the paper justified?

Yes

Should the paper be seen by a specialist statistical reviewer?

No

Do you have any concerns about statistical analyses in this paper? If so, please specify them explicitly in your report.

No

It is a condition of publication that authors make their supporting data, code and materials available - either as supplementary material or hosted in an external repository. Please rate, if applicable, the supporting data on the following criteria.

Is it accessible?

No

Is it clear?

No

Is it adequate?

No

Do you have any ethical concerns with this paper?

No

Comments to the Author

Here the authors set to examine the role of bacteriocins produced by *P. aeruginosa* on bacterial communities in differential spatial scales. The authors have shown that even though *P. aeruginosa* is known to be a ubiquitous microorganism, it is not ubiquitously found in the 70 households sampled in this study an interesting observation. The authors tested the pairwise inhibition effects of pyocins between 54 *P. aeruginosa* isolates. They found that spatial scale impacts the pyocin inhibitory effect, and the susceptibility and inhibitory effects are significantly variable between the 9 houses that samples were collected. Using whole-genome sequencing and MLST analysis, the authors found that the maximum inhibitory effects are between isolates with intermediate genetic distance.

Here are some of my concerns:

In general, I find this manuscript and the findings quite interesting. However, I feel a lack of a detailed introduction to different types of bacteriocins produced by *P. aeruginosa* makes it difficult to understand the rationale and aim of this study. I am not sure what type of pyocins are studied here; however, I assume the authors focused on R-type pyocins; some clarification is

needed.

The authors used whole-genome sequencing to analyze the genetic distance between these isolates, but no data is provided on this analysis. The authors need to include the raw sequencing reads in the supplementary information.

Minor comments:

Pyocins are called compounds, while they are protein structures

Method section: line 176 please use the correct symbol microliter

For spot assay, clarification is needed. There are no mentions on preparing the overlay plates, and it reads as the producer and indicator clones/isolates were grown in liquid cultures.

Was the pyocin extraction performed on colonies grew on plates? Moreover, how the pyocins were extracted? At the moment, this section is unclear.

Lines 188-195: the statement on confirming that zone of inhibition is due to pyocin production other than phage can be moved to the results section.

I suggest including a summary of the percentages of inhibition in three spatial scales in a table alongside the figure 1.

I suggest introducing the labeled used for households in the figure legend when it is first shown.

Change the title of the results section to the findings of that section; at the moment, it does not describe the findings.

Despite the efforts to show the effect of metabolic dissimilarity on pyocin inhibition, I find it difficult to see the rationale behind testing this, as susceptibility to pyocins is mainly due to the lack of immunity protein by different strains.

Discussion:

Discussion can be summarized, and some of the rationales for experimental plans can be moved to the result section. The authors discuss the role of pyocins in limiting the intra-species diversity of *P. aeruginosa*; however, this topic is not discussed in the introduction.

Decision letter (RSPB-2020-1706.R0)

19-Aug-2020

Dear Miss Mojesky:

Your manuscript has now been peer reviewed and the reviews have been assessed by an Associate Editor. The reviewers' comments (not including confidential comments to the Editor) and the comments from the Associate Editor are included at the end of this email for your reference. As you will see, the reviewers and the Editors have raised some concerns with your manuscript and we would like to invite you to revise your manuscript to address them.

When submitting your revision please upload a file under "Response to Referees" - in the "File Upload" section. This should document, point by point, how you have responded to the reviewers' and Editors' comments, and the adjustments you have made to the manuscript. We

require a copy of the manuscript with revisions made since the previous version marked as 'tracked changes' to be included in the 'response to referees' document.

Research ethics:

Use of animals and field studies:

It is a condition of publication that you make available the data and research materials supporting the results in the article. Please see our Data Sharing Policies (<https://royalsociety.org/journals/authors/author-guidelines/#data>). Datasets should be deposited in an appropriate publicly available repository and details of the associated accession number, link or DOI to the datasets must be included in the Data Accessibility section of the article (<https://royalsociety.org/journals/ethics-policies/data-sharing-mining/>). Reference(s) to datasets should also be included in the reference list of the article with DOIs (where available).

Online supplementary material will also carry the title and description provided during submission, so please ensure these are accurate and informative. Note that the Royal Society will not edit or typeset supplementary material and it will be hosted as provided. Please ensure that

the supplementary material includes the paper details (authors, title, journal name, article DOI). Your article DOI will be 10.1098/rspb.[paper ID in form xxxx.xxxx e.g. 10.1098/rspb.2016.0049].

Please submit a copy of your revised paper within three weeks. If we do not hear from you within this time your manuscript will be rejected. If you are unable to meet this deadline please let us know as soon as possible, as we may be able to grant a short extension.

Best wishes,
Professor Hans Heesterbeek
mailto:proceedingsb@royalsociety.org

Associate Editor
Board Member: 1

Comments to Author:

Your manuscript has been assessed by two expert reviewers. Both are positive about the work, as am I, which reveals some interesting patterns of the spatial ecology of bacterial inhibitory interactions. The reviewers make suggestions to improve the clarity and presentation of the manuscript that I would ask you to carefully consider.

Reviewer(s)' Comments to Author:

Referee: 1

Comments to the Author(s)

In this study, the authors investigate the inhibition/susceptibility profiles of *P. aeruginosa* isolates taken from different drains in different houses. Whilst I started looking at the sampling rationale with some scepticism, having read the ms, I enjoyed the thinking behind it. There is very little that needs improving in this manuscript, beyond a few typos and the comments below:

L178 - The description of the inhibition assay is a little confusing - why are the producer colonies killed prior to overlay, do the producers not need to be alive to interact with the indicator strains? This may be my ignorance, but it is not clear.

L200 - why was more genetic information not used not used for genetic distance rather than MLSTs. How many housekeeping genes were used?

L389 - Whilst I understand the data used to draw the conclusions about local adaptation, I am unsure whether there is sufficient evidence to suggest that the adaptation is to the drain environment. If this was the case a key hypothesis would be that carbon source utilisation would be significantly different (see next comment), maybe this reduction in inhibition is due to adaptation to the community, rather than the environment. As these communities are stable (a guess?), then there could have been a clonal expansion of a competitive *P. aeruginosa* that has undergone adaptive radiation perhaps...

L406 - I am a big fan of Biolog EcoPlates, however, I'm not sure that these were the best assay to measure the CLPP of these systems as they are biased towards carbon sources derived from plants rather than whatever maybe coming down the bathroom sink. I feel this section and the previous section are intertwined and maybe slimmed and combined.

Figures 3 & 4 are not a big issue, but I find them difficult to interpret as they are not clearly supporting the text. Is there a better way of illustrating this? Perhaps scatterplots with all the data?

Figure 5 is not cited in the text line 325?

Referee: 2

Comments to the Author(s)

Here the authors set to examine the role of bacteriocins produced by *P. aeruginosa* on bacterial communities in differential spatial scales. The authors have shown that even though *P. aeruginosa* is known to be a ubiquitous microorganism, it is not ubiquitously found in the 70 households sampled in this study an interesting observation. The authors tested the pairwise inhibition effects of pyocins between 54 *P. aeruginosa* isolates. They found that spatial scale impacts the pyocin inhibitory effect, and the susceptibility and inhibitory effects are significantly variable between the 9 houses that samples were collected. Using whole-genome sequencing and MLST analysis, the authors found that the maximum inhibitory effects are between isolates with intermediate genetic distance.

Here are some of my concerns:

In general, I find this manuscript and the findings quite interesting. However, I feel a lack of a detailed introduction to different types of bacteriocins produced by *P. aeruginosa* makes it difficult to understand the rationale and aim of this study. I am not sure what type of pyocins are studied here; however, I assume the authors focused on R-type pyocins; some clarification is needed.

The authors used whole-genome sequencing to analyze the genetic distance between these isolates, but no data is provided on this analysis. The authors need to include the raw sequencing reads in the supplementary information.

Minor comments:

Pyocins are called compounds, while they are protein structures

Method section: line 176 please use the correct symbol microliter

For spot assay, clarification is needed. There are no mentions on preparing the overlay plates, and it reads as the producer and indicator clones/isolates were grown in liquid cultures.

Was the pyocin extraction performed on colonies grew on plates? Moreover, how the pyocins were extracted? At the moment, this section is unclear.

Lines 188-195: the statement on confirming that zone of inhibition is due to pyocin production other than phage can be moved to the results section.

I suggest including a summary of the percentages of inhibition in three spatial scales in a table alongside the figure 1.

I suggest introducing the labeled used for households in the figure legend when it is first shown.

Change the title of the results section to the findings of that section; at the moment, it does not describe the findings.

Despite the efforts to show the effect of metabolic dissimilarity on pyocin inhibition, I find it difficult to see the rationale behind testing this, as susceptibility to pyocins is mainly due to the lack of immunity protein by different strains.

Discussion:

Discussion can be summarized, and some of the rationales for experimental plans can be moved to the result section. The authors discuss the role of pyocins in limiting the intra-species diversity of *P. aeruginosa*; however, this topic is not discussed in the introduction.

Author's Response to Decision Letter for (RSPB-2020-1706.R0)

See Appendix A.

RSPB-2020-1706.R1 (Revision)

Review form: Reviewer 1

Recommendation

Accept as is

Scientific importance: Is the manuscript an original and important contribution to its field?

Good

General interest: Is the paper of sufficient general interest?

Good

Quality of the paper: Is the overall quality of the paper suitable?

Good

Is the length of the paper justified?

Yes

Should the paper be seen by a specialist statistical reviewer?

No

Do you have any concerns about statistical analyses in this paper? If so, please specify them explicitly in your report.

No

It is a condition of publication that authors make their supporting data, code and materials available - either as supplementary material or hosted in an external repository. Please rate, if applicable, the supporting data on the following criteria.

Is it accessible?

No

Is it clear?

N/A

Is it adequate?

N/A

Do you have any ethical concerns with this paper?

No

Comments to the Author

The authors have amended the manuscript to take into account all of my points from the previous round of reviews.

Review form: Reviewer 2

Recommendation

Accept as is

Scientific importance: Is the manuscript an original and important contribution to its field?

Good

General interest: Is the paper of sufficient general interest?

Good

Quality of the paper: Is the overall quality of the paper suitable?

Good

Is the length of the paper justified?

Yes

Should the paper be seen by a specialist statistical reviewer?

No

Do you have any concerns about statistical analyses in this paper? If so, please specify them explicitly in your report.

No

It is a condition of publication that authors make their supporting data, code and materials available - either as supplementary material or hosted in an external repository. Please rate, if applicable, the supporting data on the following criteria.

Is it accessible?

Yes

Is it clear?

Yes

Is it adequate?

Yes

Do you have any ethical concerns with this paper?

No

Comments to the Author

I appreciate that the authors made excellent efforts to improve this manuscript.

Decision letter (RSPB-2020-1706.R1)

12-Oct-2020

Dear Miss Mojesky

I am pleased to inform you that your manuscript entitled "Spatial structure maintains diversity of pyocin inhibition in household *Pseudomonas aeruginosa*" has been accepted for publication in Proceedings B.

Open Access

Paper charges

Sincerely,

Professor Hans Heesterbeek

Associate Editor:

Board Member: 1

Comments to Author:

The reviewers are satisfied with the revisions, as am I. Thank you for your excellent contribution to PRSB!

Board Member: 2

Comments to Author:

(There are no comments.)

Appendix A

Dear Dr. Heesterbeek,

We thank you for handling our manuscript # RSPB-2020-1706 (Mojesky & Remold. "Spatial structure maintains diversity in pyocin inhibition of household *Pseudomonas aeruginosa*"). We thank the reviewers for providing many insightful comments that have helped us clarify and improve the quality of the manuscript. With these changes, we believe the manuscript is strengthened and will be of value to the readers upon publication in Proceedings of the Royal Society B. Our line-by-line responses follow.

Sincerely,

Aubrey Mojesky (corresponding author)

Referee: 1

Comments to the Author(s)

*In this study, the authors investigate the inhibition/susceptibility profiles of *P. aeruginosa* isolates taken from different drains in different houses. Whilst I started looking at the sampling rationale with some scepticism, having read the ms, I enjoyed the thinking behind it.*

Thank you.

There is very little that needs improving in this manuscript, beyond a few typos and the comments below:

1A. L178 - The description of the inhibition assay is a little confusing - why are the producer colonies killed prior to overlay, do the producers not need to be alive to interact with the indicator strains? This may be my ignorance, but it is not clear.

Thank you for pointing out this area where greater clarification is needed. We agree that the inhibition assay methods need further explanation. The producer colonies are killed prior to overlay to avoid the observation of inhibition due to mechanisms other than pyocin production. We have added the following clarifying text to the methods section (lines 189-191).

"Producer colonies were killed prior to soft agar overlay to avoid the observation of inhibition due to a mechanism other than pyocin production, such as contact-dependent or secretion-mediated killing."

1B. L200 - why was more genetic information not used not used for genetic distance rather than MLSTs.

Response: While there are many other ways to measure genetic distance that would involve in the inclusion of more genetic information, we chose the measurement of genetic distance through multilocus sequence comparison of housekeeping genes because it speaks to the phylogenetic history of the isolates. We have addressed this choice and its rational in lines 136-140 of the introduction and we have added a discussion of alternative measures of genetic distance in lines 446-460 of the discussion:

In the Introduction:

"This is because bacteriocin production targeting very closely related isolates will not be favored, because shared phylogenetic history will cause closely related isolates to carry the

same bacteriocin immunity genes and therefore not be killed by their shared toxins. Bacteriocin-mediated competition should also be minimal when two individuals are genetically distant if, along with other traits, their resource use overlap declines over evolutionary time.”

In the Discussion:

“Because the measure of genetic distance that we used is based on sequence comparison of conserved housekeeping genes, it measures genetic similarity based on phylogenetic history, and is not mechanistically driven by selection on pyocin use. We know that bacteriocin production in *P. aeruginosa*, particularly in household isolates [6], is much more common than that of other *Pseudomonas* and bacterial species [3]. Thus, the hump-shaped relationship between this form of genetic distance and inhibition patterns could be caused by widespread use of pyocins and restricted movement in space, which shape the population genetic structure of this species. We note that other measures of genetic distance could reveal additional evolutionary dynamics affecting patterns of pyocin interactions. For example, a decline in killing at greater genetic distance can occur if more genetically distant bacteria lack the particular cell surface receptors required for bacteriocin binding [28, 33, 34]. Such patterns would be better detected using measures of genetic distance based on gene content or gene sequence of pyocin use-associated genes such as toxin, immunity, regulatory, and receptor synthesis genes, and use of such different genetic distance measures might not reveal the hump-shaped relationship that we detected with our measure.”

1C. *How many housekeeping genes were used?*

Response: Seven housekeeping genes were used (*ascA*, *aroE*, *guaA*, *trpE*, *mutL*, *nuoD*, and *ppsA*). This method was taken from a multilocus sequence typing scheme for *P. aeruginosa* that was developed by Curran et al. [25]. We have added the following clarifying text to lines 203-206 in the methods:

“Genetic distances were estimated using Illumina whole genome sequencing and ARIBA [21] which was used to extract seven housekeeping genes (*ascA*, *aroE*, *guaA*, *trpE*, *mutL*, *nuoD*, *ppsA*) used in a *P. aeruginosa* multilocus sequence typing scheme developed by Curran et al. [22].”

1D. *L389 - Whilst I understand the data used to draw the conclusions about local adaptation, I am unsure whether there is sufficient evidence to suggest that the adaptation is to the drain environment. If this was the case a key hypothesis would be that carbon source utilisation would be significantly different (see next comment), maybe this reduction in inhibition is due to adaptation to the community, rather than the environment. As these communities are stable (a guess?), then there could have been a clonal expansion of a competitive *P. aeruginosa* that has undergone adaptive radiation perhaps...*

Response: Thank you for pointing out this area where further explanation is needed. Our thinking behind local adaptation to drain type is not constrained just the carbon source utilization; instead, we were interested in any characteristic of a drain type, including abiotic or biotic features, that may lead to local adaptation of the isolates in this study. To clarify this idea, we have added text to lines 398-404 of the discussion section (see response to comment 1F below).

1E. *L406 - I am a big fan of Biolog EcoPlates, however, I'm not sure that these were the best assay to measure the CLPP of these systems as they are biased towards carbon sources*

derived from plants rather than whatever maybe coming down the bathroom sink.

Response: We agree that the measure of carbon utilization using Biolog EcoPlates may not provide a complete picture of metabolic dissimilarity for our particular study, as these isolates are likely encountering different carbon sources within sink drains. Lines 428-432 now read:

“Additionally, the carbon resources in Biolog EcoPlates are biased toward plant-based carbon sources, which are likely different from the carbon sources available in sink drains. Future work addressing metabolic similarity could explore other resources over which competition might be occurring such as alternative carbon sources, nitrogen sources, and other bacterial nutritional requirements that might be more available to isolates in sink drains.”

1F. *I feel this section and the previous section are intertwined and maybe slimmed and combined.*

Based on this comment and comment 1D we see that the text needs language to clarify the nature of this distinction between these two sections. In the revision we aim to emphasize that the study of drain type and inhibition and the proposal of local adaptation to drain type refers not only to differences in resource availability, but other abiotic and biotic factors unique to each drain type. This has been clarified in the discussion section with the following text (lines 399-404):

“While we have not explored the particular ecological differences between bathroom and kitchen sink drains, there may be both abiotic and biotic components of these drain types that could contribute to local adaptation of *P. aeruginosa* isolates. For example, there could be differences in available resources or in microbial community compositions, either of which could affect selection through competitive interactions among *P. aeruginosa*. Future studies in this area could work to clarify the differences in drain types with respect to both abiotic and biotic factors.”

1G. *Figures 3 & 4 are not a big issue, but I find them difficult to interpret as they are not clearly supporting the text. Is there a better way of illustrating this? Perhaps scatterplots with all the data?*

Response: Thank you for making this point. We agree that these figures and the text accompanying these figures could use clarification. The purpose of figure 3 was mainly to display the stochastic variability across houses. We now convey this information by combining the section in the Results about house with the section on scale and giving the new section the title “Inhibition varies across space.” This allows us to remove figure 3 from the manuscript entirely, and instead use figure 1 to highlight the variability among houses. This is described in the following text in the results (lines 285-289):

“We found significant variability attributable to both the house of the producer and the house of the indicator. In other words, whether or not a producer is able to inhibit an indicator depends on the house of isolation for both interacting isolates (figure 1). Therefore, these results suggest, in addition to scale, there is random stochasticity across space that contributes to diversity in pyocin phenotype.”

We have also clarified the biological interpretation of figure 4 (now figure 3) by adding the following text to the results (lines 298-306):

“We found that the inhibition observed was strongly affected by the particular combination of drain types from which the producer and indicator isolates were obtained (interaction effect, $p=0.05$). In particular, when indicators from kitchen sink drains were challenged with producers from bathroom sink drains, significantly less inhibition occurred in comparison to interactions between any other combination of isolation sources, and none of the other combinations resulted in differences in inhibition (figure 3). The detection of overall greater capacity to inhibit and greater resistance to being inhibited among kitchen sink isolates (producer main effect, $p=0.0248$, indicator main effect, $p=0.0026$) is likely driven by this single combination of drain types.”

1H. *Figure 5 is not cited in the text line 325?*

Response: This figure is now cited in the text.

Referee: 2

Comments to the Author(s)

*Here the authors set to examine the role of bacteriocins produced by *P. aeruginosa* on bacterial communities in differential spatial scales. The authors have shown that even though *P. aeruginosa* is known to be a ubiquitous microorganism, it is not ubiquitously found in the 70 households sampled in this study an interesting observation. The authors tested the pairwise inhibition effects of pyocins between 54 *P. aeruginosa* isolates. They found that spatial scale impacts the pyocin inhibitory effect, and the susceptibility and inhibitory effects are significantly variable between the 9 houses that samples were collected. Using whole-genome sequencing and MLST analysis, the authors found that the maximum inhibitory effects are between isolates with intermediate genetic distance.*

Here are some of my concerns:

2A. *In general, I find this manuscript and the findings quite interesting. However, I feel a lack of a detailed introduction to different types of bacteriocins produced by *P. aeruginosa* makes it difficult to understand the rationale and aim of this study.*

Response: Thank you for this comment. Indeed, a description of pyocin diversity is critical background information that will help readers to fully understand the motivation behind this study. To provide more information on the diversity and types of pyocins, we have added the following text to the introduction (lines 77-82):

“For example, *P. aeruginosa* has the ability to produce three different types of pyocins: S, R, and F. S-type pyocins are similar to the bacteriocins of *Escherichia coli*, called colicins, and are typically composed of a toxin and immunity gene. On the other hand, R and F type pyocins resemble phage tails and resistance is thought to be mediated through receptor incompatibility [11]. Furthermore, each of these classes has multiple subtypes and isolates have the ability to produce unique combinations of each type [1, 11, 12].”

2B. *I am not sure what type of pyocins are studied here; however, I assume the authors focused on R-type pyocins; some clarification is needed.*

Response: The pyocins examined in this study included all types of pyocins: S, R, and F-type pyocins. To clarify, the following text has been added to the introduction (lines 84-87):

“The observation of inhibition in these assays can be a result of the production of any type of pyocin (S, R, and/or F) and indicates that processes occurring in the environment contribute to the maintenance of diversity in pyocin inhibition and susceptibility.”

2C. The authors used whole-genome sequencing to analyze the genetic distance between these isolates, but no data is provided on this analysis. The authors need to include the raw sequencing reads in the supplementary information.

Response: Thank you for pointing this out. We agreed that the information of sequencing and analysis requires clarification. Although whole genome sequencing was done on these isolates, we calculated genetic distance using only the seven housekeeping genes that were extracting for multilocus sequence analysis (see response to comment 1B). The concatenated MLST sequences have now been uploaded to Dryad. To elucidate the methods used in the determination of genetic distance in this study, we have added the following text to lines 206-208:

“These genes were concatenated and pairwise genetic distances were calculated using only the seven housekeeping gene sequences and R package ape [23] on RStudio version 1.1.453 [24].”

Minor comments:

The reviewer pointed out a number of places that require clarifications. Below is how we addressed each of these issues.

2D. Pyocins are called compounds, while they are protein structures

Response: Pyocins are now referred to as protein structures in the manuscript, not compounds.

2E. Method section: line 176 please use the correct symbol microliter

Response: The correct symbol for microliter is now used in the methods section.

2F. For spot assay, clarification is needed. There are no mentions on preparing the overlay plates, and it reads as the producer and indicator clones/isolates were grown in liquid cultures. Was the pyocin extraction performed on colonies grew on plates? Moreover, how the pyocins were extracted? At the moment, this section is unclear.

Response: Upon reexamination of the methods, we agree that the pairwise inhibition assay needs more clarification for the reader. Producers were spotted onto plates and allowed to grow into colonies. These colonies were then killed using chloroform, leaving only a zone in which any inhibitory compounds produced remained in the agar. Overlays were prepared by using indicator culture to inoculate soft agar, which was then overlaid on the plate. The assay therefore detects inhibition associated only with inhibitory compounds. Pyocin inhibition has a characteristic phenotype that can be differentiated from other inhibitors such as small molecules and lytic phages, allowing us to identify those combinations of producers and indicators that result in inhibition phenotypically characteristic of pyocin inhibition. This protocol is described in detail by Fyfe et al. [22]. To clarify the inhibition assay methods, the following text has been added to lines 193-195 (see also response to comment 1A above).

“Once standardized in this way, 100 μ L of each indicator culture was used to inoculate 6 mL of soft agar, which was vortexed gently and poured over the agar surface of plates that had been spotted with killed producer cultures.”

2G. *Lines 188-195: the statement on confirming that zone of inhibition is due to pyocin production other than phage can be moved to the results section.*

Response: This section has been moved to the results portion of the paper.

2H. *I suggest including a summary of the percentages of inhibition in three spatial scales in a table alongside the figure 1.*

Response: This table has been created and has been submitted along with other tables and figures.

2I. *I suggest introducing the labeled used for households in the figure legend when it is first shown.*

Response: The legend for figure 1 has been modified to include a description of the labeling used for houses in the figure. The following text has been added to the legend for figure 1:

“Light grey boxes with letters on both axes represent the nine houses from which bathroom and kitchen sink isolates were collected.”

2J. *Change the title of the results section to the findings of that section; at the moment, it does not describe the findings.*

Response: Section headings in the results section have been modified to display the findings describe in the section (e.g. “Inhibition and scale” now reads “Inhibition varies across space”, “Inhibition and drain type” now reads “Inhibition varies with drain type”).

2K. *Despite the efforts to show the effect of metabolic dissimilarity on pyocin inhibition, I find it difficult to see the rationale behind testing this, as susceptibility to pyocins is mainly due to the lack of immunity protein by different strains.*

While it is true that pyocin immunity is mediated primarily through immunity genes and/or receptor specificity, we are interested in the ecological and evolutionary mechanisms underlying these genetic patterns. However, the reviewer has pointed out an important missing link in the explanation of the reasons for looking at metabolic dissimilarity. That missing link is the role of natural selection due to metabolic overlap on resulting patterns of immunity over evolutionary time. The following text provides the explanation in the introduction (lines 122-128):

“This is because while costly production of bacteriocins is not favored when niche overlap is minimal, it becomes more beneficial as niche overlap increases. As niche overlap becomes even stronger, the capacity to inhibit the competitor may decrease not because there is insufficient benefit to bacteriocin-mediated inhibition, but because the competitors may be similar in their bacteriocin genotypes as well as their metabolic profiles, causing them to lack the capacity to inhibit one another [12, 14].”

Discussion:

2L. *Discussion can be summarized, and some of the rationales for experimental plans can be moved to the result section.*

Response: Redundancies in the discussion and text that would fit better in the results have been removed.

2M. *The authors discuss the role of pyocins in limiting the intra-species diversity of P. aeruginosa; however, this topic is not discussed in the introduction.*

Response: Thank you for pointing this out. Indeed, we believe that the role of pyocins limiting intra-species diversity at a local scale is an important finding to highlight in this study and should be presented in the introduction. The following text has been added to the introduction (lines 95-97):

“We also expect little inhibition among isolates collected within the same drain, due to the elimination of susceptible phenotypes by pyocin producers or the evolution of resistance to pyocins encountered at the within-drain level.”